# Unveiling the Role of Metabolites from a Bacterial Endophyte in Mitigating Soil Salinity and Reducing Oxidative Stress

**DOI:** 10.3390/molecules30081787

**Published:** 2025-04-16

**Authors:** Pramod Kumar Sahu, Krishna Nanda Dhal, Nakul Kale, Vivek Kumar, Niharika Rai, Amrita Gupta, Durgesh Kumar Jaiswal, Alok Kumar Srivastava

**Affiliations:** 1ICAR-National Bureau of Agriculturally Important Microorganisms, Kushmaur, Maunath Bhanjan 275103, UP, India; dhalkrishna.2016@gmail.com (K.N.D.); shivajiputra1@gmail.com (N.K.); rainiharika89@gmail.com (N.R.); amritasoni90@gmail.com (A.G.); aloksrivastva@gmail.com (A.K.S.); 2Department of Botany, Udit Narayan P. G. College, Padrauna, Kushinagar 274304, UP, India; vivek.biotech87@gmail.com; 3Department of Biotechnology, Graphic Era (Deemed to be University), Dehradun 248002, UK, India

**Keywords:** microbial metabolites, LC-HRMS, salinity, oxidative stress, endophyte

## Abstract

Several plant-associated microbes have the capability of ameliorating the adverse effects of salinity stress in plants. Such microbes produce metabolites, including proline, glycine betaine, and secondary compounds, like melatonin, traumatic acid, and *β*-estradiol, which have been found to have a role in reducing salinity-induced damage in plant cells. While the effects of these metabolites have been studied, their application-related aspects remain underexplored. In this study, we investigated the salinity-stress-alleviating potential of metabolites derived from the endophytic bacterium *Bacillus safensis* BTL5. The microbial metabolites were extracted using the hexane–chloroform fraction method and identified through LC-HRMS analysis. Four metabolites (traumatic acid, *β*-estradiol, arbutin, and *α*-mangostin), along with a fifth compound, melatonin, were initially screened for their salinity alleviation potential. Subsequently, two metabolites, i.e., arbutin and *β*-estradiol, were evaluated for their impact on growth parameters and enzymatic antioxidant activities under 200 mM salt stress. The results revealed that arbutin and *β*-estradiol significantly improved plant growth, chlorophyll content, and enzymatic activities while reducing oxidative damage. The dose-dependent effects highlighted optimal concentrations for maximum efficacy from these compounds under elevated salinity. This study signifies the potential of microbial metabolites in enhancing crop resilience to salinity, highlighting their role in sustainable agriculture. The outcomes of this study provide a baseline for the applied use of such microbial metabolites under field conditions.

## 1. Introduction

Salinity, the excessive accumulation of salts in agricultural soil, poses a significant threat to crop production worldwide [1]. It adversely affects plant growth and development by altering physiological processes, mainly causing osmotic stress, ion toxicity, oxidative stress, nutrient imbalance, changes in leaf photochemistry, reduced stomatal aperture, and impaired carbon metabolism, ultimately leading to decreased photosynthetic rates [2,3]. These combined effects can result in decreased crop yields, poor quality, and even plant death, thereby posing a serious risk to food security and agricultural sustainability. However, microbe-mediated sustainable amelioration of salinity stress has been widely reported in various crops [4,5,6].

Bio-stimulant microbes are crucial in ameliorating salinity by supplementing the plant’s inner physiological mechanisms to cope with adverse effects and improve stress resilience. These microbes employ diverse mechanisms to reduce salt-induced damage. One key mechanism is the regulation of ion homeostasis, where bio-stimulants modulate the expression of sodium and potassium transporter genes, thereby facilitating selective uptake in plants. This selective uptake balances the Na+/K+ ratio in cells and reduces sodium accumulation in tissues. Some of the key ion transporters are *SOS*1, *NHX*1, and *HKT*1 [2]. Bio-stimulants also induce the production of plant growth hormones, such as indole-3-acetic acid (IAA), gibberellins, and cytokinins. These modulations in phytohormones promote root growth and development, along with improving water and nutrient uptake under stress. A fine-tuned modulation occurs by the production of ACC deaminase, which modulates ethylene levels by breaking down ACC into *α*-ketobutyrate and ammonia, thereby regulating ethylene signaling and reducing stress in plants [7,8].

Microorganisms secrete a diverse array of stress-alleviating metabolites that can be used to mitigate salinity stress. These metabolites, including hormones, enzymes, and secondary metabolites, enhance plant growth, nutrient uptake, and stress tolerance. The applied use of these microbial metabolites holds significant potential in alleviating abiotic stresses in crop plants. Several new metabolites have been reported as having an effective role in alleviating abiotic stress in crop plants [9,10]. Interestingly, some of these stress-mitigating compounds are also produced by certain endophytes [11]. The stress-alleviating mechanisms of metabolites, like proline, glycine betaine, trehalose, and exopolysaccharides, have been studied in detail. Microbial compounds, like traumatic acid, *β*-estradiol, melatonin, arbutin, and *α*-mangostin, have been reported to have some effects related to abiotic stress alleviation. However, detailed mechanisms for salinity alleviation and their application methods have remained underexplored.

In the case of melatonin, relatively more work has been undertaken to understand its role in salinity stress. Gupta et al. [5] found that external melatonin application reduced the effects of salinity and had a synergistic impact when combined with microbial agents in rice plants. Askari et al. [12] also reported that melatonin improves photosynthetic efficiency and reduces oxidative stress under elevated salinity when applied externally. In the past decade, the molecular profiling of metabolites has identified various stress-related secondary metabolites that contribute to stress mitigation. Assessing all the metabolites secreted by microbes could provide insights into the intricate metabolic networks within microbial cells. These compounds can be targeted to expand and utilize the metabolic arsenal of microorganisms to develop climate-resilient strategies for food and nutritional security.

However, despite earlier significant advancements, several research gaps remain in understanding their mechanisms of action and optimal application strategies. Addressing these research gaps is crucial for unlocking the full potential of microbe-based metabolites in sustainable agriculture and ensuring food security in the face of increasing salinity stress. This study aims to investigate the mechanisms and optimal doses of stress-alleviating metabolites in mung beans under saline conditions.

## 2. Results and Discussion

### 2.1. LC-HRMS Analysis

The LC-HRMS analysis of the endophytic bacterium *Bacillus safensis* BTL5 culture identified 1217 metabolites (Appendix A). The total ion chromatogram (TIC) in Figure 1 shows distinct peaks corresponding to different metabolites. Additionally, the peaks of some stress alleviator compounds taken for this study are shown in Figure 2 and Figure 3.

### 2.2. Preliminary Screening of Metabolites in Mung Bean Plants

The study on preliminary screening of metabolites in the mung bean plants revealed significant findings regarding plant growth under various treatments, as detailed in Table 1. Arbutin and *β*-estradiol performed well and were thus chosen for the dose optimization study.

### 2.3. Optimizing Dosage Through the Secondary Screening of Metabolites

#### 2.3.1. Effect on Plant Growth

The effects on plant growth were recorded among the treatments differing in the concentration of two metabolites (Figure 4). Among the treatments, *β*-estradiol at 75 ppm resulted in a significantly higher fresh weight of 0.31 g. This was followed by arbutin at 50 ppm, which recorded a fresh weight of 0.28 g. In contrast, the control treatment yielded a fresh weight of 0.17 g, while arbutin at 25 ppm showed the lowest fresh weight of 0.12 g, indicating a pronounced reduction in fresh weight at this lower concentration. The shoot length data further supported these compounds’ effectiveness. Plants with arbutin at 75 ppm had a shoot length of 16.34 cm, and those with *β*-estradiol at 25 ppm reached 17.03 cm, both significantly higher than the other treatments. The control plants exhibited a lower shoot length of 12.37 cm, indicating minimal growth without treatment, while the control and arbutin plants at 25 ppm were statistically comparable. In the plant roots, both arbutin and *β*-estradiol at 50 ppm resulted in significantly higher root lengths of 6.03 cm and 5.93 cm, respectively, showing that they are effective in promoting root growth (Figure 4). However, arbutin at 75 ppm resulted in a significantly lower root length of 4.57 cm, suggesting that higher concentrations of arbutin may inhibit root development. The control plants had a mean root length of 5.37 cm, which was less than that observed with arbutin and *β*-estradiol at 50 ppm.

Additionally, the chlorophyll content indicated that both *β*-estradiol (75 ppm) and arbutin (50 ppm) were effective in enhancing chlorophyll levels and could potentially improve photosynthetic capacity. The control treatment had the lowest chlorophyll content of 23.67 SPAD units, demonstrating a limited impact on chlorophyll synthesis compared to the treated plants, all of which showed a significantly higher chlorophyll content. Overall, these results show that *β*-estradiol and arbutin can help increase the fresh weight, shoot length, root length, and chlorophyll content in mung bean plants. This result suggests a valuable role of *β*-estradiol and arbutin in agricultural practices to improve plant growth under salinity.

#### 2.3.2. Effect on Antioxidant Enzymes

The effects of different treatments on superoxide dismutase (SOD) were (Figure 4) evaluated in the mung bean seedlings. The arbutin 25 ppm plants (992.00 U mg^−1^ FW) had significantly higher SOD activity, whereas the arbutin 50 ppm (275.33 U mg^−1^ FW) and *β*-estradiol 25 ppm (260.33 U mg^−1^ FW) plants had the lowest SOD activity. The control plants had 729.33 U mg^−1^ FW SOD activity, which was significantly lower than arbutin at 25 ppm, the highest recorded value among all. The plants with 50 ppm arbutin showed showed a significantly lower SOD activity of 275.33 U mg^−1^ FW than all the other treatments. The treatments with *β*-estradiol showed variation depending upon the dose. At 25 ppm, *β*-estradiol resulted in low activity, comparable to arbutin at 50 ppm.

The effect of the different treatments on phenylalanine ammonia-lyase (PAL) activity is summarized in Figure 5. The control group showed the baseline PAL activity of 815.40 µmol TCA g^−1^ FW. The application of arbutin (25 ppm) resulted in the highest PAL activity (961.46 µmol TCA g^−1^ FW), which was significantly higher than the control. Arbutin (50 ppm) and arbutin (75 ppm) resulted in reduced PAL activity compared to 25 ppm. All *β*-estradiol treatments (25 ppm, 50 ppm, and 75 ppm) reduced PAL activity compared to the control. Among the *β*-estradiol treatments, PAL activity increased with concentration (25 < 50 < 75 ppm).

The control plants demonstrated the highest peroxidase activity, with a significantly higher value of 29.12 U mg^−1^ FW, as compared to all the other treatments. A dose-dependent increase in peroxidase activity was observed for arbutin, peaking at 75 ppm with a value of 27.27 U mg^−11^ FW. In contrast, all concentrations of *β*-estradiol resulted in similar peroxidase activity levels, which were lower than the control and arbutin at 75 ppm.

The data in Figure 5 illustrate the catalase (CAT) activity under different treatments. The control plants (380.28 U mg^−1^ FW) and the arbutin 50 ppm plants (364.83 U mg^−1^ FW) displayed the highest CAT activity, in significant contrast to the *β*-estradiol treatments. The arbutin treatments (25 and 50 ppm) resulted in intermediate CAT activity that was higher than the *β*-estradiol treatments and lower than the control. The *β*-estradiol 25 ppm (281.80 U mg^−1^ FW) and *β*-estradiol 75 ppm (301.38 U mg^−1^ FW) treatments, in contrast, resulted in significantly lower values than the others, highlighting the importance of these findings.

#### 2.3.3. Level of ROS Accumulation in Leaf

Treatment with secondary metabolite compounds at varying concentrations reduced superoxide accumulation, as indicated by the NBT staining, and influenced H_2_O_2_ accumulation, as observed through the DAB staining (Figure 6). The lowest superoxide accumulation was observed in the plants treated with 75 ppm of arbutin and 75 ppm of *β*-estradiol. In comparison, the highest accumulation was noted in the control plants and those treated with 50 ppm of arbutin. The DAB staining also revealed a similar pattern.

### 2.4. Discussion

This study was carried out to decipher the metabolic mechanisms of the salinity-alleviating endophytic bacteria *Bacillus safensis* BTL5 [2]. The LC-HRMS untargeted metabolomics approach was used to identify stress-alleviating compounds produced by *Bacillus safensis* BTL5. The metabolic profile of *B. safensis* BTL5 revealed the presence of multiple bioactive compounds with antioxidants, osmo-protectants, and growth-promoters, which might explain its stress-alleviating potential.

The untargeted metabolomics detected several polyphenolic and flavonoid derivatives, which are known for their antioxidant properties [13]. The presence of these compounds in *B. safensis* BTL5 suggests a potential role in mitigating oxidative stress. Additionally, indole-3-acetic acid (IAA) and gibberellins indicated that *B. safensis* BTL5 plays an active role in plant hormonal regulation and stress alleviation [14]. These results suggest that *B. safensis* BTL5 has the potential to sustainably improve crop growth under stress, which aligns with an earlier report on *B. subtilis* [15]. The LC-HRMS results also revealed several new metabolites that could open a promising avenue for further research on plant stress alleviation.

There were two plant trials in this study. In the first trial, four of the stress-alleviating compounds (arbutin, *β*-estradiol, traumatic acid, and *α*-mangostin) were tested in mung bean (*Vigna radiata*) plants along with melatonin. In the second trial, a dose optimization study was conducted. Evaluating endophytic stress-alleviating metabolites for salt stress tolerance in mung bean (*Vigna radiata*) plants revealed differential efficacy among the tested compounds. The arbutin and *β*-estradiol treatments worked best at 50 ppm, significantly improving the seedling growth parameters, including root length, fresh weight, and chlorophyll content, probably due to their antioxidant properties and role in mitigating salt-induced oxidative damage [16,17]. However, applying traumatic acid, melatonin, and *α*-mangostin did not improve plant growth. The lack of improvement could be attributed to suboptimal concentration levels, limited bioavailability, or an inability to effectively modulate stress-responsive pathways at 50 ppm, as observed in previous studies where some growth inducer metabolites required either higher or lower concentrations to elicit significant stress-alleviating effects [18,19]. Further investigations into concentration-dependent responses and molecular mechanisms are essential for their effective application in strategies for managing crop stress.

In the secondary screening for dose optimization, two compounds, i.e., arbutin and *β*-estradiol, were used. When applied at optimal doses, the results revealed that these metabolites significantly improved plant growth and antioxidant enzymes and reduced the accumulation of superoxide and hydrogen peroxide when exposed to salinity. This study demonstrates the potential of the microbial metabolites arbutin and *β*-estradiol to improve growth and biomass accumulation in mung bean (*Vigna radiata*) plants under salinity. It is well known that salinity disrupts plant growth through ionic toxicity, osmotic stress, and oxidative damage, resulting in reduced biomass. Applying the selected metabolites significantly alleviated these effects, promoting plant growth and enhancing biomass production, similar to melatonin in mitigating salt stress (150 mM NaCl) and enhancing biomass through cellular osmotic adjustment [5].

The reduction in oxidative stress indicates the endophyte’s ability to modulate a plant’s enzymatic and non-enzymatic antioxidant machinery. This protects cellular components from oxidative damage and sustains photosynthetic efficiency and overall plant vigor under stress. These findings align with previous studies highlighting the role of microbial metabolites and induced systemic resistance in mitigating oxidative damage [20]. The levels of reactive oxygen species were assessed by NBT and DAB staining. The NBT staining showed a noticeable drop in blue formazan precipitation in the treated plants, indicating lower superoxide levels. Such observations were reported by Lee et al. [21] in *Arabidopsis thaliana* in melatonin-treated plants. This evidence suggests that the microbial inoculants effectively upregulated superoxide dismutase (SOD) activity, converting superoxide radicals into less harmful hydrogen peroxide. Similarly, the DAB staining showed a marked reduction in brown polymer deposition in treated leaves, reflecting decreased H_2_O_2_ accumulation. The enhanced activities of superoxide dismutase (SOD) and catalase (CAT) in inoculated plants likely contributed to efficient detoxification of H_2_O_2_, further corroborating these findings. These results are similar to the findings of Zhou et al. [22] in *Solanum lycopersicum* and Nahar et al. [23] in *Vigna radiata* by glutathione (GSH) treatment.

Similar research by Tabssum et al. [24] indicated that the application of proline increases the level of superoxide dismutase (SOD) and catalase (CAT), helping in ROS scavenging under salt-stress conditions. The mode of action of these secondary metabolite compounds differed, suggesting their complementary roles within the plants, ultimately contributing to enhanced stress tolerance. Therefore, precisely applying these metabolites is crucial for maximizing their beneficial effects.

Arbutin, a phenolic glycoside, significantly improved root and shoot biomass under salinity, probably due to its protective effects as an antioxidant. These properties enable arbutin to scavenge reactive oxygen species (ROS) and alleviate oxidative stress, as evidenced by reduced levels of superoxide and hydrogen peroxide in the treated plants. These results align with the effects of arbutin application in wheat, as reported by Zhang et al. [16]. Arbutin is reported to enhance a plant’s antioxidant defense system, improve nutrient uptake, maintain osmotic balance, and stabilize cellular membranes to support metabolic processes [25]. The highest efficacy was observed at 50 ppm, significantly improving physiological and biochemical parameters compared to the other concentrations. This moderate concentration likely facilitated an optimal balance between antioxidant enhancement and metabolic stability, thus improving seedling vigor. In contrast, 25 ppm may have been insufficient for a robust response, while 75 ppm could have caused metabolic feedback inhibition or toxicity, reducing its efficacy. Overall, arbutin at 50 ppm effectively mitigated oxidative stress from salinity, improving chlorophyll content and seedling growth.

The H_2_O_2_ and superoxide staining revealed that the plants treated with arbutin experienced higher oxidative stress. This could have caused the higher enzymatic antioxidant scavenging activity shown in Figure 3. However, the plants treated with arbutin, especially at 50 ppm, could accumulate higher biomass despite higher oxidative stress. The effect on plant phytohormone status may be studied further to determine any inductive effect of arbutin under salinity. This kind of effect was reported for the inoculation of *Bacillus safensis* BTL5 in tomato [2].

*β*-estradiol, a steroid hormone, significantly enhanced plant vigor by regulating growth promotion and inducing systemic tolerance. It stimulated cell division and elongation, increasing root and shoot length and biomass while improving adaptation under osmotic stress, probably due to ionic balance. Research suggests that natural compounds, including *β*-estradiol, can affect crops differently, with responses varying across species and even genotypes. For example, glycine betaine application has shown a genotype-specific impact in wheat [26]. In the present study, the maximal effectiveness of *β*-estradiol was observed at 75 ppm. This concentration would have provided optimal stimulation for maintaining cellular homeostasis, improving stress resilience, and enhancing antioxidant defense mechanisms. At 75 ppm, *β*-estradiol contributed to increased fresh seedling weight, root length, and chlorophyll retention under salinity, suggesting a dose-dependent role in alleviating stress and promoting physiological parameters. Lower concentrations might not have been sufficient to activate the key signaling pathways necessary for stress mitigation. Additionally, other extreme concentrations of such metabolites should also be tested for their stress tolerance ability, as the standard concentrations in our current dose–response analysis could be further enhanced. Though our findings provide a basis for future studies to explore a wider range of metabolite concentrations, the specific concentration range could be further investigated.

Several studies have shown that these secondary metabolites can help plants cope with abiotic stress. Zhang et al. [16] found that arbutin improved salt tolerance in wheat by boosting antioxidant enzyme activity and reducing lipid damage. Similarly, Ahmed et al. [17] reported that *β*-estradiol improved drought resistance in maize, enhancing photosynthesis and root development. These findings suggest that secondary metabolites, like arbutin and *β*-estradiol, can enhance stress tolerance in various plant species. These metabolites also regulate stress-responsive genes and improve photosynthesis, supporting their role in stress mitigation [25,26]. Studies with arbutin application on tomato seedlings showed increased proline levels and reduced electrolyte leakage under salinity, indicating improved osmotic adjustment and membrane stability [27]. Patel et al. [28] found that *β*-estradiol improved growth and photosynthesis in rice under drought stress, indicating its role in hormone regulation and oxidative stress control. Li et al. [29] reported that combining arbutin and *β*-estradiol in soybean plants boosted root biomass and chlorophyll content, indicating synergy. Similarly, Gonzalez et al. [30] also showed that *β*-estradiol increased antioxidant enzyme activity in cucumber plants, reinforcing its role in oxidative stress regulation. These studies suggest that secondary metabolites like arbutin and *β*-estradiol improve plant stress resilience. In a nutshell, the differential response of mung bean seedlings to arbutin and *β*-estradiol at varying concentrations signifies the role of precise dose optimization for secondary metabolites in stress management strategies. While arbutin was most effective at a moderate concentration (50 ppm), *β*-estradiol required a higher concentration (75 ppm) to elicit a significant protective effect. Though the findings of this study are encouraging, further experimentation is needed to assess the bioavailability and penetration of arbutin and *β*-estradiol in plant tissues. More research is needed to understand how these metabolites interact with the key enzymes involved in stress response pathways. A gene expression analysis is also required to identify the specific genes that are differentially regulated by arbutin and *β*-estradiol.

These findings provide valuable insights into the tailored application of these metabolites for enhancing salinity tolerance in mung bean plants and potentially other crop species. Further investigations on molecular mechanisms and field evaluations would be beneficial for developing metabolite-based bio-stimulant formulations.

## 3. Materials and Methods

### 3.1. Extraction and Preparation of Crude Extracts from Bacterial Endophyte BTL5

Crude extracts of the potential bacterial endophyte *Bacillus safensis* BTL5 (Accession No. NAIMCC-B-02221) have demonstrated salinity-stress-alleviating properties, as evidenced in our previous research [2]. Briefly, the endophytic isolate was aseptically inoculated in a nutrient broth medium (125 mL) and incubated at 30 °C with continuous shaking at 150 rpm (REMI CIS-24 PLUS incubator shaker, Mumbai, India). For extraction, molecular-biology-grade solvents are used. After 1-week post-incubation, the culture broth was mixed with n-hexane in a ratio of 1:1 (*v*/*v*). This mixture was added to a separating funnel and shaken vigorously for 10 min. The mixture was allowed to settle, and each phase was settled and collected separately in a fresh collector. The same steps were repeated using a chloroform solution. Ultimately, the collected extract was air-dried in a dark place and gently scraped. The dried extract was stored at 4 °C in an amber-colored tube for further processing.

### 3.2. Untargeted Metabolomics Using LC-HRMS Analysis

#### 3.2.1. Sample Preparation for LC-HRMS

For the LC-HRMS study, the sample was prepared by mixing 100 mg of *Bacillus safensis* BTL5 crude extract in 1.5 mL of solvent (methanol and HPLC-grade water in the ratio of 80:20) and homogenized at 750 rpm for 30 min using an Eppendorf Thermo-mixer C (Eppendorf, Hamburg, Germany) at 25 °C. The mixture was then centrifuged at 3500 rpm for 10 min at 25 °C. The supernatant was filtered using a 0.22 µm syringe filter. From this, a 5 µL sample was injected into the C18 RP-HPLC column (Hypersil GOLD™, Waltham, MA, USA: particle size 1.9 µm × 2.1 mm × 100 mm). The reversed-phase chromatographic separation was performed in a gradient of solutions ranging from 0% ethanol to 95% ethanol phase in 0.1% formic acid.

#### 3.2.2. LC-HRMS Data Acquisition

The three technical replicates of the bacterial endophyte BTL5 crude extract were used for total metabolomics analysis. A Thermo Scientific Tribrid High-Resolution Accurate Mass Spectrometer “Orbitrap Eclipse” (Sunnyvale, CA, USA), coupled with an Ultra-High-Pressure Liquid Chromatography system (DionexUltiMate 3000 RSLC; Sunnyvale, CA, USA) and a Heated Electrospray Ionization (HESI) source, was employed for sample analysis following chromatographic separation. The Orbitrapanalyzer operated at a resolution of 60,000 in both positive and negative polarity modes, covering a mass range of *m*/*z* 100–1000. The instrument settings included a 35% RF lens, a 25% normalized AGC target, and an intensity threshold of 2.0 × 10^5^ for MS-OT (Master Scan). For ddMS2 OT HCD analysis, the parameters were configured as follows: quadrupole isolation mode with an isolation window of 1.5 *m*/*z*, HCD as the activation type, collision energies of 30%, 45%, and 60%, and an Orbitrap resolution of 15,000. We also set the normalized AGC target to 20%.

#### 3.2.3. Data Analysis

The mass analyzer’s raw data were processed using the “Compound Discoverer 3.3.2.31” workflow, which identifies differences between samples, predicts elemental compositions, fills gaps, and hides chemical background noise. The databases mzCloud (https://www.mzcloud.org/; accessed on 5 April 2024) and ChemSpider (https://www.chemspider.com/; accessed on 5 April 2024) were used to determine the compounds, and similarity searches were performed. We applied the mzLogic algorithm to rank the results from ChemSpider in order. Metabolism was used to map compounds to biological pathways, and batch normalization was performed based on QC. Differential analysis was calculated using *t*-test and ANOVA, and the true-to-type identification was performed using the *m*/*z* ratio in Metabolomics Workbench (https://www.metabolomicsworkbench.org; accessed on 10 April 2024) to obtain 1217 compounds.

### 3.3. Preliminary Screening of Metabolites in Mung Bean Plants

We screened the compounds from LC-HRMS analysis for different categories of functions, such as stress alleviators, antibiotics, and antioxidants. Among these compounds, four major compounds were identified and obtained in pure form by HPLC analysis. Their bioefficacy in plant growth was directly tested. These four major compounds are directly or indirectly involved in stress management, including traumatic acid, *β*-estradiol, arbutin, and *α*-mangostin (another compound, melatonin, was used as a reference to study the stress alleviation mechanism). Positive and negative controls were tested with and without salt stress, respectively. At the initial stage, 50 ppm concentrations of each compound were prepared separately, and eight grams of mung bean seeds were primed for four hours after surface sterilization aseptically. Before seed priming, the seeds were washed thoroughly with sterile distilled water, followed by a dip in 70% ethanol for one minute and a wash with sterile distilled water. Then, these primed seeds were kept for germination in a hydroponics solution under controlled conditions at room temperature (28 ± 2 °C) and relative humidity of 65–70% in a plant growth chamber. After two days of seed germination, a salt stress of 200 mM was applied physically with a Hoagland solution. The physiological parameters were measured up to eight days of growth, which included shoot length, fresh weight, and chlorophyll content.

### 3.4. Secondary Screening of Metabolites for Dose Optimization

#### 3.4.1. Effect on Plant Growth

The first plant trial showed that arbutin and *β*-estradiol at 50 ppm could have distinct effects on plant growth. Thus, a second plant trial was performed with different doses, viz., 25 ppm, 50 ppm, and 75 ppm of both compounds, along with a control (total of seven treatments: T1—control, T2—arbutin 25 ppm, T3—arbutin 50 ppm, T4—arbutin 75 ppm, T5—*β*-estradiol 25 ppm, T6—*β*-estradiol 50 ppm, and T7—*β*-estradiol 75 ppm). Like the first plant trial, eight grams of seeds were surface-sterilized with 70% ethanol and primed for four hours with the respective treatments. These seeds were left at room temperature to germinate in a hydroponics chamber. Following two days of germination, 200 mM of salt stress (NaCl) was added. After eight days of growth, the data were recorded.

The SPAD-502 Plus meter (Konica Minolta, Tokyo, Japan) is widely used in plant biology research because it is a quick, affordable, and non-destructive way to measure the amount of chlorophyll in leaves [31]. It measures the transmittance of leaves using an electromagnetic spectrum. The transmittance results are converted into a relative SPAD-502 plus meter value (0–99) proportionate to the sample’s chlorophyll concentration. In this experiment, the third leaf from the top of each plant was used to measure the chlorophyll content using the SPAD-502 Plus meter [32].

#### 3.4.2. Effect on Antioxidant Enzyme Activity

The plant extract was made from freshly harvested plants. Approximately 50 mg of fresh leaves from each treatment were homogenized in 12 mL of ice-cold phosphate buffer (100 mM, pH 7.0, containing 0.5 mM EDTA and 1.4 mmol L^−1^ mercaptoethanol) in a sterile pestle and mortar. The resulting homogenate was centrifuged at 12,000 rpm for 15 min at room temperature, and the supernatant was collected for enzyme assay analysis [33].

In the phenylalanine ammonia-lyase (PAL) assay, the reaction mixture was made from 0.2 mol L^−1^ of phosphate buffer (pH 8.7), 0.1 mol L^−1^ of phenylalanine (pH 8.7), and 1.3 mL of distilled water. Then, 0.2 mL of enzyme extract was added to start the reaction. The absorbance was measured after 30 min of incubation at 32 °C using a UV–Visible spectrophotometer (UV-1700 Pharma Spec, Shimadzu, Kyoto, Japan) at 290 nm wavelength in replicates by the addition of 0.5 mL trichloroacetic acid (1 mol L^−1^) for stopping the reaction process [34]. A mol of trans-cinnamic acid (TCA) per g of formic acid was used to express the PAL activity.

In the superoxide dismutase (SOD) assay, the mixture, which totaled 3 mL, included 200 μL of enzyme extract, 2.25 mM NBT (Nitroblue Tetrazolium), 1.5 mM sodium carbonate, 100 mM phosphate buffer (pH 7.8), and 3 mM EDTA. The process was started by adding 400 μL of riboflavin (2 mol L^−1^) to the tubes and placing them under two 15 W fluorescent lights for 15 min. We measured the SOD enzyme activity by determining the enzyme extract’s ability to stop the photochemical reduction of nitroblue tetrazolium chloride (NBT). To prevent the reaction, tubes were kept in complete darkness, and the optical density was measured at 560 nm in a UV–Visible spectrophotometer (UV-1700 Pharma Spec, Shimadzu, Kyoto, Japan) [35].

In the peroxidase (PO) assay, a reaction mixture, including 50 μL of an enzyme, 1.5 mL of pyrogallol (0.05 mol L^−1^), and 500 μL of H_2_O_2_ (1% *v*/*v*), was used to assess peroxidase activity. The change in absorbance was taken at regular intervals of 30 s up to 3 min at 420 nm in a UV–Vis spectrophotometer (UV-1700 Pharma Spec, Shimadzu, Kyoto, Japan) [36].

The catalase activity was assessed with the rate of H_2_O_2_ decomposition, measured as a change in absorbance at 240 nm using a UV–Vis spectrophotometer (UV-1700 Pharma Spec, Shimadzu, Kyoto, Japan). The reaction mixture consisted of 100 mM sodium phosphate buffer (pH 7.0), 30 mM H_2_O_2_, and 100 μL of enzyme extract in a total volume of 3.0 mL, following the method described by Aebi [37]. One unit (U) of catalase activity is the amount of enzyme required to change the absorbance of 0.001 per minute under the standard assay conditions.

#### 3.4.3. Accumulation of Reactive Oxygen Species

The accumulation of ROS in the mung leaves was studied using the method followed by Sahu et al. [38]. Briefly, H_2_O_2_ accumulation in the mung leaves was detected by 3,3-diaminobenzidine (DAB) staining, and the superoxide radical accumulation was detected by nitroblue tetrazolium (NBT) staining. The leaves were dipped in DAB and NBT solutions overnight at 37 °C. After chlorophyll removal, a solution containing ethanol, glacial acetic acid, and glycerol (3:1:1; *v*/*v*) developed dark brown and blue spots, respectively.

### 3.5. Statistical Analysis

The in planta experiments were conducted in a completely randomized design, and Duncan’s multiple range test at *p* ≤ 0.05 was used to compare the means. In the graphs, the bars with similar bars indicate non-significant differences, and the error bars show standard deviations. While DMRT is suitable for pairwise comparisons, future studies could employ multivariate analyses or bioinformatics modeling to explore complex interactions and signaling pathways.

## 4. Conclusions

Our findings contribute to the understanding of the potential of microbial metabolites as an alternative sustainable and eco-friendly strategy for alleviating salinity stress in mung beans and other crops. The findings of this study indicate that more research is needed to investigate the synergistic effects of different metabolites, validate molecular pathways, optimize application methodologies, and understand the long-term impacts on soil health and crop productivity. Gene expression studies of salt-stress-responsive pathways are also required. This study also highlights the need for more pilot trials to assess the scalability of these metabolites and their compatibility with current agronomic practices in pulses and other crop systems. However, future studies need to test other concentrations and elucidate the specific molecular pathways modulated by arbutin and *β*-estradiol in response to salt stress to further widen the mechanistic base for stress tolerance. Such investigations will be crucial for effectively integrating microbial metabolites into sustainable agricultural practices.

## Figures and Tables

**Figure 1 molecules-30-01787-f001:**
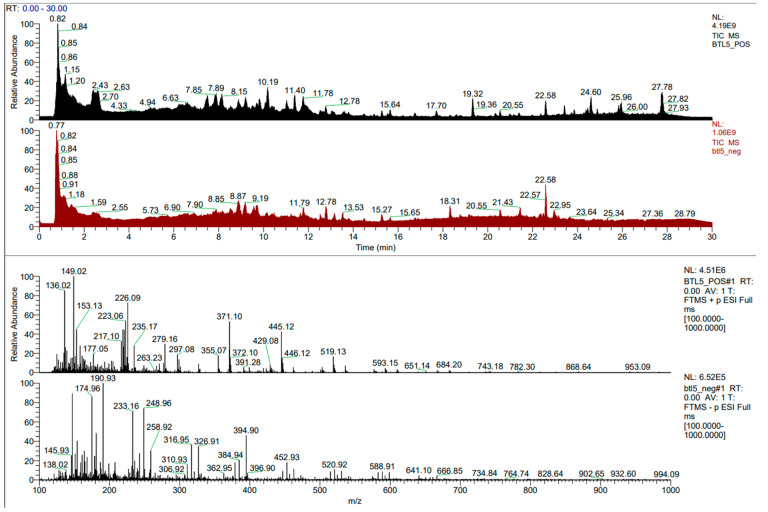
Total ion chromatogram (TIC) of the metabolites produced by *Bacillus safensis* BTL5.

**Figure 2 molecules-30-01787-f002:**
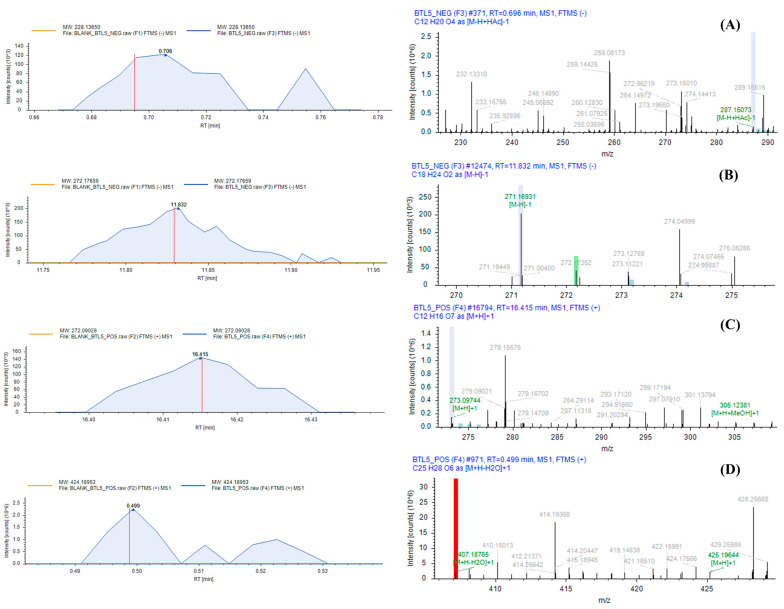
LC-HRMS identification of stress alleviator molecules present in *Bacillus safensis* BTL5: (**A**) traumatic acid, (**B**) *β*-estradiol, (**C**) arbutin, and (**D**) *α*-mangostin.

**Figure 3 molecules-30-01787-f003:**
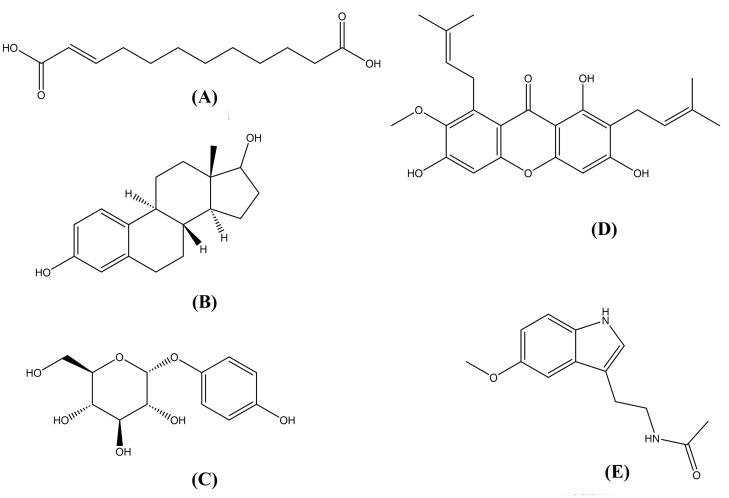
The structures of the molecules present in *Bacillus safensis* BTL5: (**A**) traumatic acid, (**B**) *β*-estradiol, (**C**) arbutin, (**D**) *α*-mangostin, and (**E**) melatonin (taken as a reference molecule for studying in planta effects).

**Figure 4 molecules-30-01787-f004:**
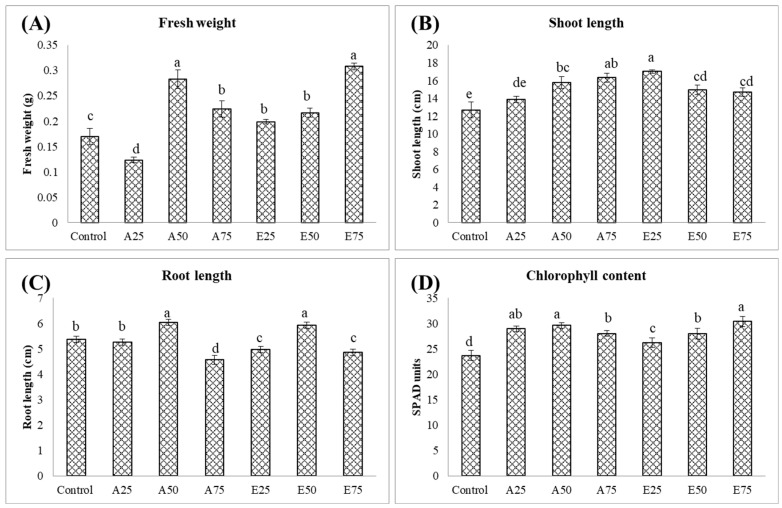
Effect of inoculation of different concentrations of arbutin and *β*-estradiol on biomass accumulation and total chlorophyll content in mung bean plants under salinity. (**A**) Fresh weight (g), (**B**) shoot length (cm), (**C**) root length (cm), and (**D**) total chlorophyll content. Error bars represent mean ± SD, *n* = 3. Treatments: (T1) control, (T2) A25 = arbutin (25 ppm), (T3) A50 = arbutin (50 ppm), (T4) A75 = arbutin (75 ppm), (T5) E25 = *β*-estradiol (25 ppm), (T6) E50 = *β*-estradiol (50 ppm), and (T7) E75 = *β*-estradiol (75 ppm). The bars with the same letter indicate a statistically non-significant difference, with ‘a’ being the highest and subsequent letters indicating chronologically lower values. The error bars show standard deviation (*p* ≤ 0.05).

**Figure 5 molecules-30-01787-f005:**
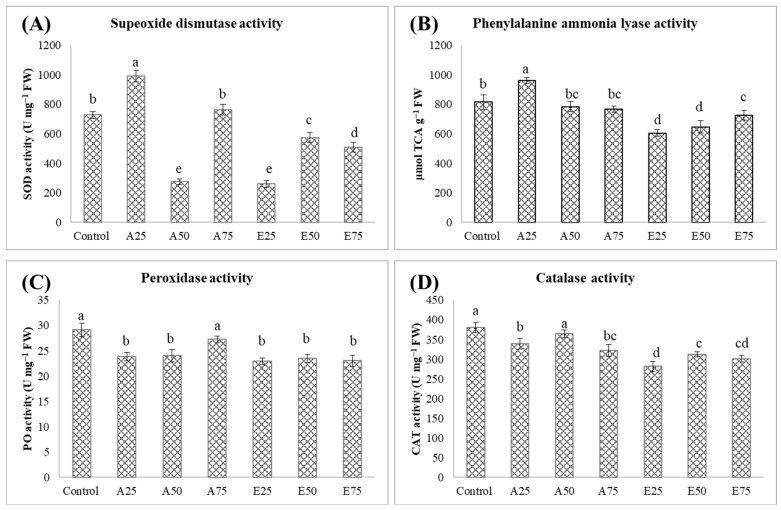
Effect of inoculation of different concentrations of arbutin and *β-estradiol* on the enzymatic antioxidants of mung bean plants under salinity: (**A**) superoxide dismutase (SOD), (**B**) phenylalanine ammonia-lyase (PAL), (**C**) peroxidase (PO), and (**D**) catalase activity (CAT). Treatments: (T1) control, (T2) A25 = arbutin (25 ppm), (T3) A50 = arbutin (50 ppm), (T4) A75 = arbutin (75 ppm), (T5) E25 = *β-estradiol* (25 ppm), (T6) E50 = *β-estradiol* (50 ppm), and (T7) E75 = *β-estradiol* (75 ppm). The bars with the same letter indicate a statistically non-significant difference, with ‘a’ being the highest and subsequent letters indicating chronologically lower values. The error bars show standard deviation (*p* ≤ 0.05).

**Figure 6 molecules-30-01787-f006:**
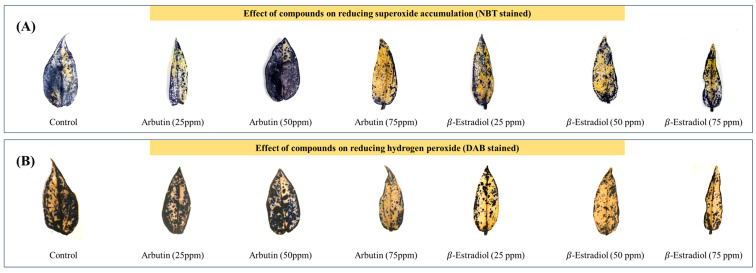
Effect of compounds on reducing (**A**) superoxide accumulation (NBT stained) and (**B**) hydrogen peroxide (H_2_O_2_) accumulation (DAB stained) in the different treatments of mung bean plants.

**Table 1 molecules-30-01787-t001:** The effect of different metabolites found in *Bacillus safensis* BTL5 on mung bean growth parameters.

Sl No.	Treatment	Fresh Weight (g)	Shoot Length (cm)	Chlorophyll Content (SPAD Units)
1	Negative Control	0.17 ^de^	13.60 ^b^	27.33 ^b^
2	Positive control	0.15 ^f^	8.77 ^e^	22.77 ^cd^
3	Traumatic acid	0.16 ^ef^	9.70 ^d^	24.17 ^c^
4	*β*-Estradiol	0.24 ^a^	13.63 ^b^	26.87 ^b^
5	Melatonin	0.20 ^c^	11.73 ^c^	26.63 ^b^
6	Arbutin	0.22 ^b^	15.60 ^a^	29.67 ^a^
7	*α*-Mangostin	0.18 ^d^	9.23 ^de^	18.10 ^e^
8	Endophyte (BTL5)	0.18 ^d^	11.53 ^c^	21.93 ^d^

Note: Means are separated by DMRT, mean values with similar letters do not differ significantly at *p* ≤ 0.05.

## Data Availability

The data are contained within this article and the Appendix A.

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
