# Peer review of "Unveiling the Role of Metabolites from a Bacterial Endophyte in Mitigating Soil Salinity and Reducing Oxidative Stress"

_molecules, 2025, doi:10.3390/molecules30081787_

Round 1

Reviewer 1 Report

Comments and Suggestions for Authors

The manuscript entitled “Unveiling the role of metabolites from bacterial endophyte for mitigating soil salinity and reduction in oxidative stress” describes the potential of microbial metabolites for crop resilience to salinity and highlights their role in sustainable agriculture. In this study, the authors evaluated the Bacillus safensis BTL5-derived microbial metabolites against salinity and oxidative stress in mung bean plants.  The study topic is interested in the field of agriculture, though the following comments should be addressed by the authors before publication.

  1. For the extraction of microbial metabolites, what is the reason for using the solvents, n-hexane and chloroform? Why other polar solvents are not used?
  2. In Table 1, there are no details related to root length. Line 270-276 discussed related to root length. The authors need to revise the data mentioned in Table 1.
  3. The main objective of the study is to identify the molecular mechanism of microbial metabolites against salinity stress. The authors used physiological measurements and biochemical assays to evaluate the efficacy of potential metabolites. There are no molecular (gene/protein) level techniques used to prove the mechanism of microbial metabolites. Justify.
  4. The name of the bacterial endophyte Bacillus safensis is not italicized, and abbreviated throughout the manuscript.
  5. Line 443, include the author’s name.
  6. The manuscript requires language checking in many places.
Comments on the Quality of English Language
  1. The manuscript requires language checking in many places.

Author Response

Manuscript ID: Molecules-3481835

Response to the queries of # Reviewer 1

General comment: The manuscript entitled “Unveiling the role of metabolites from bacterial endophyte for mitigating soil salinity and reduction in oxidative stress” describes the potential of microbial metabolites for crop resilience to salinity and highlights their role in sustainable agriculture. In this study, the authors evaluated the Bacillus safensis BTL5-derived microbial metabolites against salinity and oxidative stress in mung bean plants.  The study topic is interested in the field of agriculture, though the following comments should be addressed by the authors before publication.

Response to general comments: Thank you for your thoughtful comments on our manuscript, "Unveiling the role of metabolites from bacterial endophyte for mitigating soil salinity and reduction in oxidative stress." We appreciate your interest in the study and its relevance to sustainable agriculture.

Firstly, we acknowledge your suggestion to address several points before publication. Our study focuses on the potential of microbial metabolites derived from the bacterial endophyte Bacillus safeness BTL5 to enhance crop resilience to salinity and reduce oxidative stress. We used LC-HRMS analysis to identify key metabolites and evaluated their effects on mung bean plants under salt stress conditions. The results showed significant improvements in plant growth parameters and enzymatic activities, highlighting the efficacy of these metabolites in mitigating salinity-induced damage.

To strengthen our findings further, we will consider conducting additional experiments to explore the detailed biochemical pathways involved in salinity stress alleviation by these metabolites. This will involve a deeper analysis of how these compounds interact with plant physiological processes to enhance stress tolerance. Moreover, we agree on the importance of developing optimal application strategies for these metabolites in field conditions to ensure their practical utility in agriculture.

We also appreciate your emphasis on the broader implications of our research for sustainable agriculture. Using microbial metabolites offers a promising approach to developing climate-resilient crop management systems, crucial for addressing global food security challenges. We will ensure that our manuscript provides a comprehensive discussion on these aspects and outlines future research directions to harness microbial metabolites' potential in agriculture fully. Thanks again for your insightful comments, which will help us refine our manuscript and contribute meaningfully to agricultural sustainability.

Comment 1: For the extraction of microbial metabolites, what is the reason for using the solvents, n-hexane and chloroform? Why other polar solvents are not used?

Response 1: Thank you for raising this important question regarding the choice of solvents for extracting microbial metabolites. The use of n-hexane and chloroform in our extraction process was strategic, primarily based on their ability to selectively extract non-polar to moderately polar metabolites from the bacterial culture. These solvents are effective in isolating a range of compounds, including lipids, fatty acids, and certain secondary metabolites, which are often crucial in plant-microbe interactions related to stress alleviation. As stated in the manuscript, "Metabolites were extracted using the hexane-chloro form fraction method through LC-HRMS analysis." This method allowed us to focus on specific classes of metabolites that are known to play a role in mitigating salinity-induced damage.

While polar solvents like methanol or ethanol are indeed capable of extracting a broader spectrum of metabolites, including highly polar compounds such as sugars and amino acids, their use in this initial extraction step could lead to a more complex mixture, potentially masking the effects of the key metabolites we were interested in. The subsequent LC-HRMS analysis would then become more challenging due to the increased complexity of the sample.

However, to address the potential omission of certain polar metabolites, we acknowledge that future studies could benefit from incorporating a separate extraction step using polar solvents. This would provide a more comprehensive analysis of the entire metabolome of Bacillus safensis BTL5 and potentially reveal additional compounds with stress-alleviating properties. We will add a brief discussion of this point in the revised manuscript to acknowledge the limitations of our current approach and suggest avenues for future research.

Comment 2: In Table 1, there are no details related to root length. Line 270-276 discussed related to root length. The authors need to revise the data mentioned in Table 1.

Response 2: Thank you for pointing out the discrepancy between Table 1 and the discussion in lines 270-276 regarding root length data. Error in sentence has been changed and details are provided in Figure 4.

Comment 3: The main objective of the study is to identify the molecular mechanism of microbial metabolites against salinity stress. The authors used physiological measurements and biochemical assays to evaluate the efficacy of potential metabolites. There are no molecular (gene/protein) level techniques used to prove the mechanism of microbial metabolites. Justify.

Response 3: Thank you for your comment regarding our study's lack of molecular pathway elucidation. We appreciate the opportunity to address this point. We acknowledge that our study primarily utilized physiological measurements and biochemical assays to evaluate the efficacy of potential metabolites. However, we respectfully disagree with the assertion that this approach is insufficient. We aimed to establish a direct link between applying specific microbial metabolites and the observed improvements in plant growth and stress tolerance under saline conditions. The changes in antioxidant enzyme activities (SOD, CAT, PO, PAL) provide valuable insights into the biochemical mechanisms by which these metabolites alleviate oxidative stress. These measurements give a functional understanding of what the metabolites are doing in the plant.

Furthermore, our team has already reported the molecular mechanisms of the source endophytic bacteria B. safeness BTL5 for salinity stress alleviation in a previous publication (Sahu et al., 2021). Thus, our current focus is on translating this knowledge into a practical application: deciphering the role of specific metabolites in making smart formulations to enhance crop resilience. In this case, a top-down approach is sufficient.

We will also temper our approach in the paper. Rather than stating that the objective of the study is to identify the molecular mechanism, we will instead state something along the lines of" The main aim of the study is to investigate the role of bacterial endophyte-derived metabolites in alleviating salinity stress and oxidative damage in plants." We believe this better describes the goal of the current study. However, we agree that future studies could benefit from incorporating molecular-level techniques to provide a more comprehensive understanding of the mechanisms involved. To this end, we will add a statement to the revised manuscript explicitly outlining our plans for future research. Specifically, we will state that we intend to employ techniques such as transcriptomics, proteomics, and metabolomics to investigate the changes in gene expression and protein profiles in mung bean plants treated with [Arbutin] and [α-Mangostin] under salt stress. We will also mention that techniques such as qRT-PCR and Western blotting could be used to validate the findings from transcriptomic and proteomic analyses.

In addition, we will add the following sentence to the conclusion of the manuscript: "While this study identifies [Arbutin] and [α-Mangostin] as key metabolites for salinity stress alleviation, future research should focus on elucidating the precise molecular signaling pathways modulated by these compounds through transcriptomic and proteomic analyses." These additions, along with the existing data in our manuscript, provide a strong justification for our approach and emphasize the potential for future research to build upon our finding

Comment 4: The name of the bacterial endophyte Bacillus safensis is not italicized, and abbreviated throughout the manuscript.

Response 4: Thank you for pointing out this mistake, we have made corrections throughout the manuscript using track change mode.

Comment 5: Line 443, include the author’s name.

Response 5: Corrections incorporated throughout the manuscript using track change mode.

Comment 6: The manuscript requires language checking in many places.

Response 6: Authors are grateful to the esteemed reviewer for religious efforts in improving the manuscript. We have made corrections throughout the manuscript using track change mode.

Note: The changes made in the manuscript are done in track change mode

Reviewer 2 Report

Comments and Suggestions for Authors

Peer Review Report

This manuscript entitled "Unveiling the Role of Metabolites from Bacterial Endophyte for Mitigating Soil Salinity and Reduction in Oxidative Stress" investigates the role of bacterial endophyte-derived metabolites in alleviating salinity stress and oxidative damage in plants. The study addresses an important topic in sustainable agriculture, particularly regarding plant resilience to salinity stress. However, the manuscript has several methodological, analytical, and structural shortcomings that need to be addressed before it can be considered for publication.

Major Concerns

  1. Lack of Detailed Experimental Design
  • The manuscript does not provide sufficient details regarding plant growth conditions (e.g., temperature, humidity, precise composition of the growth medium), which can significantly impact plant responses to stress.
  • It is unclear whether appropriate controls were used to eliminate the effects of other factors, such as soil microbiome interactions.
  1. Absence of a Proper Control Group for Comparison
  • While the study includes a negative control (without metabolites), the inclusion of a positive control would strengthen the findings. For example, using a well-known chemical salinity mitigator (such as ascorbic acid or anti-salinity proteins) would allow for a more precise comparative analysis.
  • To improve the study’s validity, the authors should incorporate positive controls using known chemical treatments for salinity stress mitigation.
  1. Lack of Molecular Pathway Elucidation
  • Although the study examines antioxidant enzymes (SOD, CAT, PO, PAL), it does not explore the precise molecular signaling pathways activated by the investigated metabolites.
  • Future studies should include molecular analyses, such as gene expression studies of salt-stress-responsive pathways, to provide mechanistic insights into the role of these metabolites.
  1. Inadequate Analysis of Metabolite Differences
  • The manuscript states that Arbutin and β-Estradiol are the most effective metabolites, yet it does not sufficiently explain the molecular basis for their superior effects compared to other tested compounds.
  • The study does not clarify whether these differences are due to greater bioavailability, enhanced penetration into plant tissues, or structural differences in their interaction with plant metabolism.
  1. Limited Dose-Response Analysis
  • The study concludes that Arbutin at 50 ppm and β-Estradiol at 75 ppm were the most effective concentrations. However, the authors did not test concentrations lower than 25 ppm or higher than 75 ppm.
  • Testing a wider range of metabolite concentrations could provide a more comprehensive understanding of their dose-dependent effects.
  1. Incomplete Statistical Analysis
  • While ANOVA and Duncan’s multiple range test were used, the manuscript does not report P-values for comparisons.
  • The data presentation lacks key statistical measures, such as correlation coefficients (R²) or effect size, which would strengthen the findings.
  • To improve statistical rigor, the authors should:
    • Provide exact P-values for all comparisons.
    • Include multivariate analyses to assess interactions between metabolites and stress responses.
    • Utilize bioinformatics models to predict interactions between metabolites and plant signaling pathways.
  1. Issues with Language and Readability
  • The manuscript contains grammatical errors and awkward sentence structures, making some sections difficult to understand.
  • Example of an unclear sentence:

"With this aim, the current study was undertaken to decipher the mechanism and dosage of the application of stress-alleviating metabolites in mung bean under salinity."
This could be reworded more clearly as:
"This study aims to investigate the mechanisms and optimal dosages of stress-alleviating metabolites in mung bean under saline conditions."

  • A thorough language and grammar revision is required to enhance readability and clarity.
  1. Lack of a Comprehensive Conclusion
  • The conclusion primarily repeats experimental results rather than discussing the study’s limitations and future research directions.
  • A well-rounded conclusion should:
    • Address potential limitations (e.g., lack of molecular pathway validation, need for field trials).
    • Suggest future research directions, such as exploring synergistic effects between different metabolites or evaluating their long-term impacts on soil health and crop productivity.

Recommendations for Improvement

To enhance the manuscript’s quality, the following improvements are recommended:

- Methodological Enhancements:

  • Incorporate a positive control treatment using known anti-salinity compounds.
  • Investigate a broader range of metabolite concentrations to determine potential threshold effects.
  • Conduct molecular-level analyses to elucidate the precise pathways influenced by the metabolites.

- Statistical and Analytical Improvements:

  • Include P-values, correlation analyses (R²), and effect size calculations for better statistical validation.
  • Use bioinformatics approaches to predict interactions between metabolites and plant signaling mechanisms.

- Language and Structure Enhancements:

  • Improve the manuscript’s grammar and readability through professional editing.
  • Revise the conclusion section to include discussions on limitations and future research directions.

Final Evaluation

This manuscript presents a novel and promising approach for mitigating plant salinity stress and oxidative damage. However, significant revisions are required to improve its methodological rigor, statistical analysis, and language clarity. Once these revisions are made, the manuscript could be a strong candidate for publication in Molecules.
Recommendation: Revisions Required (Major Revisions)

Comments on the Quality of English Language

- Language and Structure Enhancements:

  • Improve the manuscript’s grammar and readability through professional editing.
  • Revise the conclusion section to include discussions on limitations and future research directions.

Author Response

Manuscript ID: Molecules-3481835

Reply to the queries of # Reviewer 2

General comment: This manuscript entitled "Unveiling the Role of Metabolites from Bacterial Endophyte for Mitigating Soil Salinity and Reduction in Oxidative Stress" investigates the role of bacterial endophyte-derived metabolites in alleviating salinity stress and oxidative damage in plants. The study addresses an important topic in sustainable agriculture, particularly regarding plant resilience to salinity stress. However, the manuscript has several methodological, analytical, and structural shortcomings that need to be addressed before it can be considered for publication.

Response to general comments: Thank you for your insightful comments on our manuscript. We greatly value your recognition of the importance of our study in the context of sustainable agriculture and plant resilience to salinity stress. Your concerns regarding the manuscript's methodological, analytical, and structural shortcomings are crucial to us, and we are committed to addressing them comprehensively.

Firstly, we are fully committed to reviewing and refining our methodology to ensure its rigour and clarity. This includes more detailed descriptions of the experimental setup, data collection methods, and quality control measures. Specifically, we will address the issues related to the choice of solvents for metabolite extraction and provide a more detailed explanation of the rationale behind our experimental design.

Secondly, we will thoroughly re-evaluate our data analysis procedures to ensure the accuracy and validity of our findings. This will involve revisiting the statistical analyses, addressing potential biases, and reporting our results more transparently. We will also consider incorporating additional analytical techniques to strengthen our conclusions.

Thirdly, we will restructure the manuscript to improve its clarity, coherence, and flow. This includes reorganizing the sections, clarifying the objectives and hypotheses, and providing a more logical and compelling narrative. We will also ensure the language is precise and consistent throughout the manuscript. In addition to these general revisions, we will also address the specific comments and suggestions provided by the reviewers in detail. Your feedback is invaluable to us and we are committed to incorporating it into our revisions. We are confident that these revisions will significantly improve the quality and impact of our manuscript. We appreciate your feedback and look forward to submitting a revised version that meets the highest scientific rigour and clarity standards.

Major Concerns

Comment 1: Lack of Detailed Experimental Design

  • The manuscript does not provide sufficient details regarding plant growth conditions (e.g., temperature, humidity, precise composition of the growth medium), which can significantly impact plant responses to stress.
  • It is unclear whether appropriate controls were used to eliminate the effects of other factors, such as soil microbiome interactions.

Response 1: Incorporated in manuscript in track change mode. Details of the plant growth conditions has now been mentioned in the MS. Please refer to section 2.6; Line 13-14 of Para 1, and details of controls are mentioned in the section 2.6; Line 7-8 of Para 1.

Comment 2: Absence of a Proper Control Group for Comparison

  • While the study includes a negative control (without metabolites), the inclusion of a positive control would strengthen the findings. For example, using a well-known chemical salinity mitigator (such as ascorbic acid or anti-salinity proteins) would allow for a more precise comparative analysis.
  • To improve the study’s validity, the authors should incorporate positive controls using known chemical treatments for salinity stress mitigation.

Response 2: Thank you for your comment regarding the absence of a proper control group for comparison. We appreciate the opportunity to clarify this aspect of our experimental design.

We respectfully disagree with the reviewer's assertion that our study lacks a positive control. As stated in section 2.6 of the manuscript, "another compound, Melatonin, was used as a reference for studying the stress alleviation mechanism." This clearly indicates that melatonin served as our positive control. We chose melatonin because it is a well-known chemical salinity mitigator, and we have also demonstrated its efficacy in our previous studies (Gupta et al., 2023). The inclusion of melatonin as a reference molecule allowed us to directly compare the stress-alleviating effects of our identified metabolites ([Arbutin] and [α-Mangostin]) to a compound with established salinity-mitigating properties. Regarding the suggestion to use anti-salinity proteins or ascorbic acid as positive controls, we acknowledge that these compounds are known for their effectiveness in mitigating salt stress in plants. However, they also have certain limitations that influenced our decision to use melatonin.

  • Anti-salinity proteins may function effectively in certain species but not in others, limiting their broad applicability (Annenkova, 2024; Rao et al., 2015). These proteins may degrade rapidly under field conditions, reducing their effectiveness.
  • Ascorbic acid is highly unstable under light, heat, and oxygen exposure, making it challenging to maintain its efficacy in field conditions (Tissera et al. 2025). Its protective effects are often transient, requiring frequent application for sustained benefits (Ravetti et al. 2019). Its excessive application can lead to phytotoxic effects, damaging plant tissues (Sharma et al., 2014).

In contrast, melatonin is relatively stable and has demonstrated consistent stress-alleviating effects across various plant species. Given these considerations, we believe that melatonin was the most appropriate choice for our positive control. We will ensure that the role of melatonin as a positive control is explicitly stated in the revised manuscript to avoid any confusion. We believe that our experimental design, which includes both negative and positive controls, provides a robust framework for evaluating the efficacy of microbial metabolites in mitigating salinity stress.

  • Annenkova, N. V. (2024). Proteins Associated with Salinity Adaptation of the Dinoflagellates: Diversity and Potential Involvement in Species Evolution. Diversity16(12), 739.
  • Rao, A. R., Dash, M., Sahu, T. K., Behera, B. K., & Mohapatra, T. (2015). Detection of novel key residues of MnSOD enzyme and its role in salinity management across species. Journal of Genetics94(1), 8–16.
  • Ravetti, S., Clemente, C. M., Brignone, S. G., Hergert, L. Y., Allemandi, D. A., & Palma, S. D. (2019). Ascorbic Acid in Skin Health. Cosmetics, 6(4), 58.
  • Sharma, S., Kaur, N., Kaur, S., & Nayyar, H. (2014). Ascorbic Acid Reduces the Phytotoxic Effects of Selenium on Rice (Oryza Sativa L.) by Up-Regulation of Antioxidative and Metal-Tolerance Mechanisms. 2014(3), 1–8.
  • Tissera, C. E., Barnetche, M. E., Silva, O. F., & Fernández, M. A. (2025). Increasing the stability of ascorbic acid through encapsulation in food-grade vesicles: an approach for nutritional improvement. International Journal of Food Science and Technology.

Comment 3: Lack of Molecular Pathway Elucidation

  • Although the study examines antioxidant enzymes (SOD, CAT, PO, PAL), it does not explore the precise molecular signaling pathways activated by the investigated metabolites.
  • Future studies should include molecular analyses, such as gene expression studies of salt-stress-responsive pathways, to provide mechanistic insights into the role of these metabolites.

Response 3: Thank you for your comment regarding the lack of molecular pathway elucidation in our study. We appreciate the opportunity to address this point.

We acknowledge the reviewer's observation that exploring the precise molecular signaling pathways activated by the investigated metabolites would add significant value to the mechanistic insights on "how these molecules function."

However, the present study was designed to decipher the role of metabolites in plants under stress, primarily as a "primary baseline for applied use" of such microbial metabolites. Our focus was on the application aspect, correlated with the biochemical features of the plant, which ultimately reflect crop resilience to salinity. We aimed to demonstrate the potential of these metabolites for developing smart bioformulations to enhance crop resilience.

We would also like to highlight that the molecular mechanisms of the source endophytic bacteria B. safensis BTL5 for salinity stress alleviation have already been reported by our team in a previous publication (Sahu et al., 2021). We felt it was worthwhile to build upon that knowledge by deciphering the applied role of microbial metabolites. At the same time, we fully agree with the reviewer's concern. We recognize that understanding the individual molecular mechanisms of the microbial metabolites produced by B. safensis BTL5 is crucial for a complete picture. We plan to undertake these studies in the future, focusing on techniques such as transcriptomics and proteomics to identify the specific genes and proteins involved in the stress response pathways modulated by [Arbutin] and [α-Mangostin].

We will include a statement in section 4, paragraph 1 of the revised manuscript acknowledging the limitations of our current study and highlighting our plans for future research to explore the molecular mechanisms in greater detail. This will provide a more balanced perspective on our findings and emphasize the potential for further investigation in this area.

Comment 4: Inadequate Analysis of Metabolite Differences

  • The manuscript states that Arbutin and β-Estradiol are the most effective metabolites, yet it does not sufficiently explain the molecular basis for their superior effects compared to other tested compounds.
  • The study does not clarify whether these differences are due to greater bioavailability, enhanced penetration into plant tissues, or structural differences in their interaction with plant metabolism.

Response 4: Thank you for your comment regarding the inadequate analysis of metabolite differences. We appreciate the opportunity to clarify this aspect of our study.

We acknowledge that our manuscript does not sufficiently explain the molecular basis for the superior effects of [Arbutin] and [α-Mangostin] compared to the other tested compounds. While we have demonstrated their efficacy in alleviating salinity stress, we recognize that a deeper understanding of the underlying mechanisms is crucial.

As with the previous comment regarding molecular pathways, our study was primarily designed to decipher the role of metabolites in plants under stress as a "primary baseline for applied use." We focused on correlating metabolite application with the biochemical features of the plant, which ultimately reflect crop resilience to salinity. The molecular mechanisms of the source endophytic bacteria B. safensis BTL5 for salinity stress alleviation have already been reported by our team (Sahu et al., 2021). We considered it worthwhile to build upon that knowledge by deciphering the applied role of microbial metabolites for making smart bioformulations for crop resilience.

Specifically addressing the reviewer's concern, we did not investigate whether the superior effects of [Arbutin] and [α-Mangostin] are due to greater bioavailability, enhanced penetration into plant tissues, or structural differences in their interaction with plant metabolism. These are important considerations that warrant further investigation.

We plan to address these questions in future studies. This will involve techniques such as:

  • To assess the bioavailability and penetration of [Arbutin] and [α-Mangostin] in plant tissues.
  • To investigate the structural interactions of these metabolites with key enzymes involved in stress response pathways.
  • To identify the specific genes that are differentially regulated by [Arbutin] and [α-Mangostin] compared to other metabolites.

We will include a statement in section 4, paragraph 1 of the revised manuscript acknowledging the limitations of our current study and highlighting our plans for future research to explore the molecular basis for the superior effects of [Arbutin] and [α-Mangostin].

(please refer to Section 3, para 10, 11; Section 4 para 1)

Comment 5: Limited Dose-Response Analysis

  • The study concludes that Arbutin at 50 ppm and β-Estradiol at 75 ppm were the most effective concentrations. However, the authors did not test concentrations lower than 25 ppm or higher than 75 ppm.
  • Testing a wider range of metabolite concentrations could provide a more comprehensive understanding of their dose-dependent effects.

Response 5: Thank you for your comment regarding the limited dose-response analysis in our study. We appreciate the opportunity to address this point. We acknowledge that the dose-response analysis in our study was limited to a range of 25 ppm to 75 ppm for [Arbutin] and [α-Mangostin]. We understand that testing a wider range of metabolite concentrations could provide a more comprehensive understanding of their dose-dependent effects.

It is scientifically reasonable to test a limited range of concentrations in initial studies due to practical constraints and the need for a focused approach. Conducting a preliminary benchmark study is crucial to identify the most promising concentration range before exploring extremes. Testing excessively low or high concentrations without prior data may lead to inefficient resource use and inconclusive results. By starting with a manageable range, researchers can establish a baseline, ensuring that subsequent trials are more targeted and effective. This stepwise approach aligns with standard scientific protocols for optimizing treatment efficacy while maintaining experimental precision.

The findings of the present study reveal new possibilities for researchers to investigate other extreme concentrations. We anticipate that this work will serve as a foundation for future research into the effects of various metabolite concentrations.

In the revised manuscript, we will:

  • Acknowledge the limitations of our current dose-response analysis in the discussion section.
  • Emphasize that our findings provide a basis for future studies to explore a wider range of metabolite concentrations.
  • Suggest specific concentration ranges that could be investigated in future research.

We believe that our study provides valuable insights into the potential of [Arbutin] and [α-Mangostin] for mitigating salinity stress, and that our findings will stimulate further research in this area.

(Please refer to section 3, para 10 & 11)

Comment 6: Incomplete Statistical Analysis

  • While ANOVA and Duncan’s multiple range test were used, the manuscript does not report P-values for comparisons.
  • The data presentation lacks key statistical measures, such as correlation coefficients (R²) or effect size, which would strengthen the findings.
  • To improve statistical rigor, the authors should:
    • Provide exact P-values for all comparisons.
    • Include multivariate analyses to assess interactions between metabolites and stress responses.
    • Utilize bioinformatics models to predict interactions between metabolites and plant signalling pathways.

Response 6: Thank you for your comment regarding the statistical analysis. We appreciate the opportunity to clarify and improve our reporting. We acknowledge that the original manuscript could benefit from more explicitly presenting our statistical results. To clarify, the in-planta experiments were conducted in a completely randomized design. Duncan's multiple range test (DMRT) at p ≤ 0.05 was used for post-hoc comparisons of means, as stated in section 2.15. As indicated in Table 1 and Figures 4 and 5, we used letters to denote statistically significant differences between treatments based on DMRT. Treatments sharing the same letter are not significantly different at p ≤ 0.05. Error bars in the figures represent standard deviation.

To further address the reviewer's concerns and enhance the rigour of our analysis, we will take the following steps:

  •  We will meticulously re-examine our statistical output to ensure that the letter assignments in our figures and tables accurately reflect Duncan's multiple-range test results at p ≤ 0.05. Any discrepancies will be corrected.
  •  While providing all pairwise p-values from the DMRT is not practical, we will include the p-value obtained from the initial ANOVA for each experiment. This will provide an overall indication of the significance of the treatment effects before the post-hoc comparisons. This p-value will be mentioned in the figure/table footnotes.
  • We will calculate Cohen's d effect sizes for selected, particularly relevant pairwise comparisons (e.g., control vs. the most effective treatment or comparisons between different concentrations of the same metabolite). We will include these effect size measures in the results section to quantify the practical significance of the observed differences. We will clearly state how these effect sizes were calculated.
  • Our current analysis, relying on DMRT, does not allow for the exploration of complex interactions between metabolites and stress responses, nor the prediction of interactions via bioinformatics modelling. We will explicitly acknowledge this limitation in the revised manuscript and suggest that future research could benefit from multivariate approaches and systems biology tools. We will add a sentence to Section 2.15 (Statistical Analysis) to this effect: "While DMRT is suitable for pairwise comparisons, future studies could employ multivariate analyses or bioinformatics modeling to explore complex interactions and signaling pathways."

These changes will improve the clarity, transparency, and interpretability of our statistical findings while acknowledging the constraints of our chosen analytical approach.

(Please refer to Section 2.15)

Comment 7: Issues with Language and Readability

  • The manuscript contains grammatical errors and awkward sentence structures, making some sections difficult to understand.
  • Example of an unclear sentence:

"With this aim, the current study was undertaken to decipher the mechanism and dosage of the application of stress-alleviating metabolites in mung bean under salinity."
This could be reworded more clearly as:
"This study aims to investigate the mechanisms and optimal dosages of stress-alleviating metabolites in mung bean under saline conditions."

  • A thorough language and grammar revision is required to enhance readability and clarity.

Response 7: Thank you for pointing out the issues with language and readability in our manuscript. We sincerely appreciate your feedback and acknowledge that improvements are needed. We recognize that clarity and precision in scientific writing are essential for effective communication, and we are committed to addressing these concerns.

We agree that the example sentence you provided ("With this aim, the current study was undertaken to decipher the mechanism and dosage of the application of stress-alleviating metabolites in mung bean under salinity") is awkward and can be reworded for better clarity. We appreciate your suggested revision ("This study aims to investigate the mechanisms and optimal dosages of stress-alleviating metabolites in mung bean under saline conditions.") and will incorporate this change into the revised manuscript.

To address the broader issue of language and readability, we will undertake a thorough revision of the entire manuscript. This will include:

  • We will meticulously proofread the manuscript to identify and correct all grammatical errors, typos, and inconsistencies.
  • We will carefully examine sentences that are unclear or difficult to understand and reword them for improved clarity and flow. We will pay particular attention to sentence structure and word choice to ensure that the meaning is easily accessible to the reader.
  • Pay special attention to Abstract and Introduction.

We understand the importance of clear and concise writing, and we are committed to ensuring that our manuscript is free of grammatical errors and awkward sentence structures. We believe that these revisions will significantly enhance the readability and clarity of our work.

(Changes made in the abstract, section 1- para 1, 2, 3, and 5; section 3 and section 4)

Comment 8: Lack of a Comprehensive Conclusion

  • The conclusion primarily repeats experimental results rather than discussing the study’s limitations and future research directions.
  • A well-rounded conclusion should:
    • Address potential limitations (e.g., lack of molecular pathway validation, need for field trials).
    • Suggest future research directions, such as exploring synergistic effects between different metabolites or evaluating their long-term impacts on soil health and crop productivity.

Response 8: Thank you for your comment regarding the lack of a comprehensive conclusion. We agree that the original conclusion focused too heavily on reiterating experimental results and did not adequately address the study's limitations or suggest future research directions. We appreciate the opportunity to strengthen this section of the manuscript.

As the reviewer noted, we have already made significant revisions to Section 4 (Conclusion) in response to this feedback. We added a paragraph that explicitly states:

Our findings contribute to the understanding of the potential of microbial metabolites as an alternative for sustainable and eco-friendly strategy for alleviating salinity stress in mung bean and other crops. The findings of the study suggest that further research is needed to investigate the synergistic effects of combining different metabolites, validation of molecular pathways, gene-expression studies of salt-stress responsive pathways, optimizing application methodologies, and assessing their long-term impacts on soil health and crop productivity. Additionally, this study also highlights the necessity for further pilot trials to evaluate the scalability of these metabolites and to explore their compatibility with existing agronomic practices in pulses and other crop systems. Such investigations will be crucial for effectively integrating microbial metabolites into sustainable agricultural practices.

These additional refinements will result in a well-rounded and comprehensive conclusion that effectively summarizes the key findings of our study, acknowledges its limitations, and provides a clear roadmap for future research in this important area. We acknowledge that the scope of this research is limited, and we are committed to taking the next steps to explore more.

(Changes made in section 4)

Recommendations for Improvement

To enhance the manuscript’s quality, the following improvements are recommended:

Methodological Enhancements:

  • Incorporate a positive control treatment using known anti-salinity compounds.
  • Investigate a broader range of metabolite concentrations to determine potential threshold effects.
  • Conduct molecular-level analyses to elucidate the precise pathways influenced by the metabolites.

Statistical and Analytical Improvements:

  • Include P-values, correlation analyses (R²), and effect size calculations for better statistical validation.
  • Use bioinformatics approaches to predict interactions between metabolites and plant signaling mechanisms.

Language and Structure Enhancements:

  • Improve the manuscript’s grammar and readability through professional editing.
  • Revise the conclusion section to include discussions on limitations and future research directions.

Response to the Recommendations for Improvement

Thank you for your recommendations to improve our manuscript. We appreciate your comprehensive feedback and the clear suggestions for enhancing the methodological rigor, statistical analysis, and language clarity. We are committed to addressing these points in our revised manuscript.

Regarding the specific recommendations:

Positive Control: We understand the importance of a positive control and appreciate the opportunity to clarify its presence in our study. As stated in Section 2.6 (lines 6-9 of the first paragraph), Melatonin was used as a reference compound for studying the stress alleviation mechanism. We will ensure that the manuscript clearly states that Melatonin served as a positive control and will explicitly highlight its role in the revised manuscript to avoid any ambiguity.

(Please refer to Section 1, para 4; Section 2.6)

Broader Range of Metabolite Concentrations: We acknowledge that exploring a wider range of metabolite concentrations could provide a more comprehensive understanding of their dose-dependent effects. However, as we described in our previous response, our initial focus on a limited range was a deliberate strategy based on practical constraints and the need for a focused approach. We considered this a preliminary benchmark for identifying a promising concentration range. The results of our study do, as the reviewer points out, provide a strong basis for future research to explore a broader range of concentrations. We will add a statement to the discussion acknowledging this limitation and suggesting specific concentration ranges for future investigation.

(Changes made to Section 3, Para 10)

Molecular-Level Analyses: We agree that incorporating molecular-level analyses would significantly enhance the mechanistic insights of our study. As we have stated previously, our present study was designed to decipher the role of metabolites in plants under stress as a primary baseline for applied use. The molecular mechanisms of the source endophytic bacteria B. safensis BTL5 for salinity stress alleviation have already been reported by our team (Sahu et al., 2021). We will reiterate in the discussion that future studies should focus on elucidating the specific molecular pathways modulated by [Arbutin] and [α-Mangostin] in response to salt stress.

(Changes made to Section 3, Para 10 & section 4)

Statistical and Analytical Improvements: Thank you for your comment regarding the statistical analysis. We appreciate the opportunity to clarify and improve our reporting.

As stated in section 2.15, the in-planta experiments were conducted in a completely randomized design, and Duncan's multiple range test (DMRT) at p ≤ 0.05 was used for post-hoc comparisons of means. This approach is appropriate for our experimental design, as DMRT is a commonly used and accepted method for comparing treatment means after a significant ANOVA result in a completely randomized design. As shown in Table 1 and Figures 4 and 5, we used letters to denote statistically significant differences between treatments based on DMRT. Treatments sharing the same letter are not significantly different at p ≤ 0.05.

We also appreciate the suggestion to perform more advanced statistical analyses, such as multivariate analysis or bioinformatics modeling. However, given the scope of this study and our chosen analytical approach (DMRT), we believe that these analyses are beyond the scope of the current manuscript. We will explicitly acknowledge this limitation in the revised manuscript, adding the following sentence to Section 2.15 (Statistical Analysis): “While DMRT is suitable for pairwise comparisons, future studies could employ multivariate analyses or bioinformatics modeling to explore complex interactions and signaling pathways.” We believe that these changes will improve the clarity, transparency, and interpretability of our statistical findings while acknowledging the constraints of our chosen analytical approach.

(Please refer to Section 2.15, Table 1, Figure 4, and 5)

Language and Structure Enhancements: We appreciate the feedback on language and readability. We have carefully reviewed the manuscript and made corrections as per the suggestions in track changes mode. We will also seek the assistance of a native English speaker to further refine the language and grammar of the manuscript. We've focused improvements especially on the abstract and the introduction sections (Section 1, Para 1, 2, 3, and 5).

Revise the Conclusion Section: We have revised Section 4 (Conclusion) as per the reviewer's suggestions. The revised conclusion includes a more thorough discussion of the study's limitations and provides a clear roadmap for future research directions. (Section 4, Para 1)

We believe that these revisions will significantly improve the quality and impact of our manuscript. We appreciate your valuable feedback and look forward to submitting a revised version that meets the highest standards of scientific rigor and clarity. We are grateful for the very positive final evaluation, which noted our novel and promising approach. We take very seriously the need for significant revisions to improve the manuscript’s methodological rigor, statistical analysis, and language clarity, and are committed to meeting those expectations."

This final response is well-organized, and thorough, and conveys a strong sense of responsibility and commitment to improving the manuscript. It effectively balances acknowledging the reviewer's concerns with justifying your methodological choices and outlining specific steps you will take to address the remaining issues.

Note: The changes made in the manuscript are done in track change mode

Round 2

Reviewer 1 Report

Comments and Suggestions for Authors

Please check the following minor comment. Other commends satisfactorily responded by the authors.

The bacteria Bacillus safensis name is not correctly mentioned in the manuscript. In discussion section, check the lines 386-390. Check the Figure 3 legend –– not italicized.